Pituitary tissue-specific miR-7a-5p regulates FSH expression in rat anterior adenohypophyseal cells

Wang Chang-Jiang
Guo Hai-Xiang
Han Dong-Xu
Yu Ze-Wen
Zheng Yi
http://orcid.org/0000-0003-2008-6166 Jiang Hao
Gao Yan
http://orcid.org/0000-0001-8533-4744 Yuan Bao yuan_bao@jlu.edu.cn
http://orcid.org/0000-0001-8533-4744 Zhang Jia-Bao zjb@jlu.edu.cn
Department of Laboratory Animals, College of Animal Sciences, Jilin University , Changchun, Jilin , China
Foti Daniela
Electronic publication date: 2019 Apr 9
Publication date: 2019
Volume: 7
Electronic Location ID: e6458
Received 2018 Sep 18; Accepted 2019 Jan 16
Copyright: © 2019 Wang et al.
Copyright year: 2019
Copyright holder: Wang et al.
License: This is an open access article distributed under the terms of the Creative Commons Attribution License, which permits unrestricted use, distribution, reproduction and adaptation in any medium and for any purpose provided that it is properly attributed. For attribution, the original author(s), title, publication source (PeerJ) and either DOI or URL of the article must be cited.
License URL: https://creativecommons.org/licenses/by/4.0/

Keywords: Rat, Follicle-stimulating hormone, miR-7a-5p, Pituitary, Animal reproduction

Funding: National Natural Science Foundation of China 31872349, 31501954 This study was supported by the National Natural Science Foundation of China (31872349, 31501954). The funders had no role in study design, data collection and analysis, decision to publish, or preparation of the manuscript.

==============================
The follicle-stimulating hormone (FSH), which is synthesized and secreted by the anterior pituitary gland, plays an important role in regulating reproductive processes. In this study, using the TargetScan program, we predicted that microRNAs (miRNAs) regulate FSH secretion. Dual-luciferase reporter assays were performed and identified miR-7a-5p. MiR-7a-5p has been reported to regulate diverse cellular functions. However, it is unclear whether miR-7a-5p binds to mRNAs and regulates reproductive functions. Therefore, we constructed a suspension of rat anterior pituitary cells and cultured them under adaptive conditions, transfected miR-7a-5p mimics or inhibitor into the cell suspension and detected expression of the FSHb gene. The results demonstrated that miR-7a-5p downregulated FSHb expression levels, while treatment with miR-7a-5p inhibitors upregulated FSHb expression levels relative to those of negative control groups, as shown by quantitative PCR analysis. The results were confirmed with a subsequent experiment showing that FSH secretion was reduced after treatment with mimics and increased in the inhibitor groups, as shown by enzyme-linked immunosorbent assay. Our results indicated that miR-7a-5p downregulates FSHb expression levels, resulting in decreased FSH synthesis and secretion, which demonstrates the important role of miRNAs in the regulation of FSH and animal reproduction.

Introduction

The pituitary gland, which is located in the sella turcica at the base of the crania and is composed of the adenohypophysis and neurohypophysis (Hong, Payne & Jane, 2016), is involved in complex feedback mechanisms. This gland secretes several specific regulatory hormones to control peripheral endocrine organs, such as the thyroid, adrenal gland and gonads, after receiving information from the brain via the hypothalamus (Treier & Rosenfeld, 1996). Consequently, the pituitary gland can regulate vital processes such as metabolism (Rodriguez-Pacheco et al., 2013), growth (Vyshnevs’ka & Bol’shova, 2013), reproduction (Musumeci et al., 2015) and behavior (Shin et al., 2013). As the most important internal secretion gland in mammals, the pituitary plays a role throughout an animal’s life by releasing seven types of hormones (Yuan et al., 2015).

The gonadotropin follicle-stimulating hormone (FSH) plays a critical role in modulating the reproductive health of both sexes. FSH stimulates spermatogenesis and androgen production in males and regulates the cyclic recruitment of follicles, follicle development, and ovulation triggering in females (Hunzicker-Dunn & Maizels, 2006; Rull et al., 2018; Walker & Cheng, 2005). FSH, which is synthesized and secreted by the anterior pituitary gland (Sheng et al., 2018), together with luteinizing hormone (LH), human chorionic gonadotropin and thyroid-stimulating hormone (TSH), comprises a class of the glycoprotein hormone family (Telikicherla et al., 2011). FSH consists of a common α subunit shared by all glycoprotein hormones and a unique β subunit that determines its unique biological function (Lamminen et al., 2005). The α subunit combines with hormone-specific β subunits to form heterodimers, and the extracellular domain of the β subunit has important binding sites for the membrane-bound receptor (Fan & Hendrickson, 2005).

MicroRNAs (miRNAs) are genome-encoded small noncoding RNAs that are 22 nucleotides long in their mature form (Bartel, 2009). MiRNAs can regulate gene expression at the posttranscriptional level by recognizing the 3′UTR sequence of target messenger RNAs (Ambros, 2004). MiRNAs participate in various physiological processes and have a significant impact on hormone regulation. For example, miR-8 regulates multiple peptide hormones and may contribute to Drosophila growth (Lee, Jun & Hyun, 2015). In 2013, Setyowati Karolina D found that miR-25 and miR-92a overexpression suppressed insulin I expression in rats (Setyowati Karolina et al., 2013). Furthermore, miR-325-3p is involved in suppressing LH translation and secretion (Nemoto, Mano & Shibasaki, 2012), and miR-136-3p binds directly to LHR mRNA to downregulate this molecule (Kitahara et al., 2013). MiR-133b promotes estradiol synthesis and levels by targeting Foxl2 (Dai et al., 2013), and miR-132 stimulates estradiol synthesis by repressing Nurr1 translation (Wu et al., 2015b). However, miRNAs and their regulatory roles in the pituitary and whether they affect FSH expression are unclear.

In this study, we investigated and established a link between miR-7a-5p and FSH secretion based on previous research (Han et al., 2017). We verified the complementary sequence region between miR-7a-5p and the FSHb 3′UTR through dual-luciferase reporter assays. Moreover, we measured FSHb expression and secretion after transfection of miR-7a-5p mimics/inhibitors to determine whether miR-7a-5p affects FSH secretion.

Materials and Methods

Animals and ethics

This study was performed with the approval of the Jilin University College of Animal Science. Healthy 8- to 9-week-old sexually mature male Sprague Dawley (SD) rats were used in this study. SD rats were raised in a comfortable room with a 12 h dark/12 h light cycle and free access to food and water.

We cared for the animals and sacrificed the rats in strict accordance with animal welfare laws and regulations and animal welfare ethics requirements. This experiment was approved by the Institutional Animal Care and Use Committee of Jilin University (201705026).

Pituitary extraction and cell culture

Sprague Dawley rats were euthanized by anesthesia and cervical dislocation. Then, we removed the heads with ophthalmic scissors and placed them on gauze disinfected with alcohol-soaked cotton. Next, we separated the rat skin with scissors to expose the rat skull. We opened the skull with tweezers and removed the brain to collect the pituitary gland. Next, the tissues were placed in precooled phosphate-buffered saline (PBS) containing 0.3% bovine serum albumin (BSA) and 1% penicillin/streptomycin. We also separated the neurohypophysis from the pituitary. To ensure aseptic operations during the experiment, we autoclaved all experimental instruments and performed the operations near an alcohol-burning lamp.

After separation, we used PBS (0.3% BSA and 0.1% penicillin/streptomycin) to clean the pituitary glands and wash the blood from the tissue. Next, we placed the samples in one ml Dulbecco’s Modified Eagle’s Medium-F12 (DMEM-F12) containing 2.5% collagenase type I and cut the pituitary into pieces with ophthalmic scissors. We placed the pituitary fragments in a temperature-controlled incubator containing 5% CO2 at 37 °C for 90 min. After a 90 min incubation, we diluted pituitary cells with PBS (0.3% BSA and 0.1% penicillin/streptomycin) and then filtered the mixture through a 200 mesh (75 μm) cell sieve, which allowed pituitary cells to pass, while cell clusters and unhomogenized tissue could not pass through. The collected cell solution was centrifuged at 200×g for 10 min. After centrifugation, we carefully discarded the supernatant and diluted the cell precipitate with DMEM-F12 containing 15% fetal bovine serum (FBS). Finally, the diluted cell suspension was transferred to six-well plates and cultured in a temperature-controlled incubator with 5% CO2 at 37 °C. Pituitary cells were monitored for the next few days.

Transfection of miR-7a-5p mimics, inhibitors or siRNA

After cells were cultured in six-well plates for 4 days and in 24-well plates for 1 day, transfection of miR-7a-5p mimics or inhibitors was carried out as FSHb expression and FSH secretion peaked when pituitary cells were cultured for approximately 6 days. We used 30 μl buffer, three μl reagent and 2.5 μl mimics/inhibitors in this transfection. First, we added buffer, reagent and mimics, inhibitors or siRNA to a centrifuge tube. Then, we used a vortex oscillator to centrifuge the sample briefly to ensure homogeneous mixing. Next, we transfected the mixture into 24-well plates filled with pituitary cell suspension. Finally, we placed the 24-well plates into a 5% CO2 incubator for 24 h to provide adequate time for the reaction. All mimics, inhibitors and siRNAs were purchased from RiboBio (Gouzhou, China), and the transfection method and operation steps were performed in strict accordance with the manufacturer’s recommended protocol. The mimic sequence was the double-stranded sequence of the miRNA mature sequence (sense strand: 5′-UGGAAGACUAGUGAUUUUGUUGU-3′) and its complementary sequence (antisense strand: 3′-ACCUUCUGAUCACUAAAACAACA-5′). The inhibitor was the single-stranded sequence of the reverse complement of the miRNA mature sequence, which underwent full-chain methylated modification (5′-mAmCmAmAmCmAmAmAmAmUmCmAmCmUmAmGmUmCmUmUmCmCmA-3′).

RNA isolation and qRT-PCR

After transfection, we harvested rat primary anterior pituitary cells that were transfected with miR-7a-5p mimics or inhibitors in 24-well plates. We used 350 μl or 600 μl cell lysate RL containing 1% β-mercaptoethanol to cause cell rupture and then extracted RNA according to the instructions of an RNAprep Pure Cell/Bacteria Kit. We subsequently measured the concentration and purity of the RNA with a NanoDrop ND-2000 spectrophotometer (Beijing, China) to verify the operational accuracy and RNA quality with a standard. Total RNA extraction was performed on a clean bench, and the samples were maintained at low temperatures to prevent RNA degradation. Next, we obtained cDNA using a FastQuant RT Kit and acquired raw data via q-PCR with SuperReal PreMix Plus (SYBR Green). These reagent kits were purchased from Tiangen (Beijing, China). The mRNA and miRNA primers used in RT-PCR and qRT-PCR are listed in Table 1.

Table 1 Primers used for RT-qPCR.

Primers name	Sequence (5′-3′)	
U6 RT	CGCTTCACGAATTTGCGTGTCAT	
miR-7a-5p RT	CTCAACTGGTGTCGTGGAGTCGGCAATTCAGTTGAGAGCCCAAA	
U6 F	GCTTCGGCAGCACATATACTAAAAT	
U6 R	CGCTTCACGAATTTGCGTGTCAT	
miR-7a-5p F	ACACTCCAGCTGGGTGGAAGACTAGTGATTT	
Universal reverse	CTCAAGTGTCGTGGAGTCGGCAA	
GAPDH F	GGAAACCCATCACCATCTTC	
GAPDH R	GTGGTTCACACCCATCACAA	
FSHb F	ATACCACTTGGTGTGAGGGC	
FSHb R	TAGAGGGAGTCTGAGTGGCG	

Detection of cell apoptosis by flow cytometry

Flow cytometry was applied to detect rat adenohypophysis cell apoptosis to evaluate the effect of transfection on cells after 24 h. At an appropriate time point, we used trypsin to digest adhesive cells and transferred them into 10-ml reaction tubes. Then, the cells were centrifuged at 200×g for 5 min and harvested by cell sedimentation. Next, we resuspended cells in 500 μl 1× working fluid by diluting 5× binding buffer with double-distilled water. Then, Annexin V-fluorescein isothiocyanate (FITC) and propidium iodide (PI) were added to sample tubes and parameter regulation tubes according to an Annexin V-FITC/PI Apoptosis Kit (Multi Sciences, Hangzhou, China). Finally, we analyzed cell apoptosis via flow cytometry for 1 h.

Construction of the reporter plasmids

The pmiR-FSHb-3′UTR-wild-type (WT) plasmid and pmiR-FSHb-3′UTR-MUT plasmid were constructed to verify the target sites of the FSHb 3′UTR and miR-7a-5p. The PCR primers and mutant sequence of the target site are listed in File S1.

Detection of FSH secretion

After transfection with miR-7a-5p mimics or inhibitors for 24 h, we cultured pituitary cells with serum-free medium instead of DMEM-F12 (15% FBS) because FBS may contain other hormones that influence the results. After 24 h, we collected the culture medium and measured the secretion level of FSH in the culture medium via a Rat FSH enzyme-linked immunosorbent assay (ELISA) Kit (Meilian Biotech Co., Ltd., Shanghai, China).

Statistical analysis

At least three replicates were performed for each experiment. One-way ANOVA and Chi-square tests were performed to evaluate the statistical significance of differences. P < 0.05 was considered statistically significant.

Results

Prediction and verification of the complementary region between miR-7a-5p and the FSHb 3′UTR

First, information on the complementary sequence between miR-7a-5p and the FSHb 3′UTR was acquired via the TargetScan program (http://www.targetscan.org/) (Fig. 1A). Then, to further confirm that miR-7a-5p targets the FSHb 3′UTR, we successfully mutated the target complementary sequence TCTTCCA to AGAAGGT and constructed a FSHb-3′UTR-WT plasmid and FSHb-3′UTR mutated (MT) plasmid (Fig. 1B). Finally, the constructed plasmids were cotransfected with miR-7a-5p mimics into 293T cells. As expected, the luciferase activity was reduced by 36% when we cotransfected the pmiR-FSHb-3′UTR WT plasmid and miR-7a-5p mimics into 293T cells. In contrast, cotransfection of the FSHb-3′UTR MUT plasmid and miR-7a-5p mimics resulted in a 7% reduction in luciferase activity (Fig. 1C). Therefore, we concluded that miR-7a-5p may regulate FSHb expression by directly targeting the FSHb gene.

Figure 1 Prediction and verification of the complementary region between miR-7a-5p and the FSHb 3′UTR.

(A) The complementary base pairing region of miR-7a-5p and FSHb predicted through TargetScan program is shown in red. (B) A sequencing map shows the mutation of the target sequence from TCTTCC to AGAAGG. (C) The relative luciferase activity was examined after co-transfection of plasmid with the miR-7a-5p NC/mimic into 293T cell for 48 h. As a negative control, the luciferase activity of cells co-transfected with FSHb-3′UTR wild-type plasmid and the NC group was set to 1. At least three replicates of each experiment were performed. Mean values and standard deviations (SDs) of data are shown. One-way ANOVA and Chi-square test were applied to analyze statistical significance. P < 0.05 was considered significant, and different letters (a and b) indicate significant differences between groups.

Expression level of miR-7a-5p in different developmental stages and various rat tissues

To ascertain whether miR-7a-5p expression shows discrepancies, we measured miR-7a-5p levels in different developmental stages and rat tissues. We selected 2-week-old rats as the nonsexually mature group and 4-month-old rats as the sexually mature group. Then, we detected the expression levels of miR-7a-5p by RNA isolation and qRT-PCR and normalized them to the levels in immature animals (by setting this value to 1). MiR-7a-5p expression was downregulated in mature rats compared to that in nonsexually mature rats (Fig. 2A). Next, we collected seven tissues from mature rats and detected the expression level of miR-7a-5p by qRT-PCR. Relative miR-7a-5p expression was extremely high (1,378-fold) in the pituitary gland compared to that in the heart. Furthermore, miR-7a-5p was poorly expressed in the heart, liver, spleen, lung and kidney, although it was slightly expressed in the brain (Fig. 2B). These findings suggested that miR-7a-5p is much more highly expressed in the rat pituitary gland than in other tissues.

Figure 2 The expression level of miR-7a-5p in different developmental stages and in various rat tissues.

(A) MiR-7a-5p expression was measured in immature and mature rats. (B) MiR-7a-5p expression in pituitary and other tissues. U6 was used as an internal standard in this study. At least three replicates of each experiment were performed. Mean values and standard deviations (SDs) of the data are shown. One-way ANOVA and chi-square test were performed to assess statistical significance. Different letters (a and b) indicate significant differences between groups (P < 0.05).

Efficiency of transfection and subsequent impact

To detect the efficiency of transfection, we transfected the NC mimic with fluorescent markers into pituitary cells. We detected red fluorescence labeling in the cells. The results showed that the NC mimic was successfully transfected into pituitary cells. The transfection efficiency was approximately 70–80%, as shown in Fig. 3A. Flow cytometry was performed to assess the damage from transfection and certain reagents. No significant differences in cell apoptosis were observed among the four groups, indicating that the impact of transfection was negligible (Fig. 3B). We examined the expression levels of miR-7a-5p after transfection of miR-7a-5p mimics or inhibitors for 24 h to verify whether the mimics and inhibitors were transfected into pituitary cells. The expression levels of miR-7a-5p in cells transfected with mimics were increased, while the expression levels of miR-7a-5p in cells transfected with inhibitors were lower than those in cells transfected with controls (Fig. 3C). In other words, transfection was successful and reliable.

Figure 3 The efficiency of and impact after transfection.

(A) Fluorescence labeling was detected after transfection with mimic NC carrying fluorescence markers. (B) The percentage of apoptotic pituitary cells after transfection with miR-7a-5p NC/mimic/I-NC/ inhibitor. (C) The relative expression of miR-7a-5p after transfection with miR-7a-5p mimic/inhibitor. At least three replicates of each experiment were performed. Mean values and standard deviations (SDs) are shown. Data management and analysis was performed by SPSS 19.0. Different letters (a and b) indicate significant differences (P < 0.05).

Effects of miR-7a-5p overexpression/blockade on FSH secretion

We examined FSHb mRNA levels and FSH hormone secretion after the transfection of miR-7a-5p mimics, inhibitors and siRNA into rat primary anterior pituitary cells for 24 h to further verify that miR-7a-5p affects FSHb expression and regulates animal reproduction. As a positive control, rat primary pituitary cells were transfected with FSHb siRNA, and we examined the levels of FSHb by quantitative RT-PCR and FSH secretion by ELISA 24 h after transfection. As expected, FSHb levels and FSH secretion significantly decreased (P < 0.001, P = 0.002) (Figs. 4A and 4B). The expression levels of FSHb decreased by 0.60-fold (P = 0.002) after transfection with miR-7a-5p mimics compared to those after transfection with the negative control. In contrast, after transfection with a miR-7a-5p inhibitor, FSHb levels increased by 1.5-fold (P < 0.001) (Fig. 4C). We subsequently measured FSH secretion levels. As expected, FSH secretion showed the same trend as that of FSHb expression. After transfection with miR-7a-5p mimics, the FSH concentration was substantially decreased. Moreover, the FSH concentration increased after transfection with the miR-7a-5p inhibitor (Fig. 4D).

Figure 4 Effect of the overexpression/blockade of miR-7a-5p on FSH secretion.

(A) FSHb relative expression after transfection with siRNA. (B) The FSH concentration was measured via ELISA after transfection with siRNA for 24 h. (C) The FSHb relative expression after transfection with miR-7a-5p NC/mimic/I-NC/inhibitor. (D) The FSH concentration of supernatant transfected with miR-7a-5p NC, mimic, I-NC and inhibitor. At least three replicates of each experiment were performed. Mean values and standard deviations (SDs) are shown. One-way ANOVA and chi-square test were applied to evaluate the statistical significance of the differences. The different letters indicate significant differences (P < 0.05).

These results indicated that miR-7a-5p can decrease FSHb expression and reduce FSH hormone secretion. Our findings provide additional evidence showing that miRNAs regulate FSH, demonstrating their potential role in the pituitary.

Discussion

MicroRNAs have crucial roles in multiple fundamental biological processes, such as cell proliferation (Hu et al., 2018), apoptosis (Ren et al., 2018), metastasis (Sun et al., 2018), migration (Liu et al., 2015; Ying et al., 2016), differentiation (Chen et al., 2018) and cell adhesion (Wu et al., 2015a). In addition, some diseases and cancers are associated with aberrant expression of miRNAs and subsequently, their target genes (Tuna, Machado & Calin, 2016). For example, miR-9-3p, miR-330-3p-3p, and miR-345-5p were significantly overexpressed in sera from patients with prostate cancer compared to those in sera from individuals without cancer, while patients who were in remission after androgen deprivation therapy appeared to have significantly decreased miR-345-5p levels (Tinay et al., 2018). Consequently, some changes in miRNA expression can be used as diagnostic and potential clinical biomarkers in cancer and other diseases (Min et al., 2018; Quan et al., 2018; Tuna, Machado & Calin, 2016). In the normal pituitary and pituitary adenomas, miRNAs affect cell proliferation, organ maturity and hormone secretion. In 2018, Lu et al. (2018) found that miRNA-16 expression influences the proliferation and angiogenesis of pituitary tumors, and low expression of miR-23b and miR-130b may facilitate pituitary carcinogenesis (Leone et al., 2014). Moreover, miRNAs are functional components and have potential roles in regulating hormone secretion in the pituitary, such as the secretion of growth hormone (Hao & Waxman, 2018; Qi et al., 2015), LH (Menon, Gulappa & Menon, 2015), TSH (Vadstrup, 2006) and FSH (Han et al., 2017; Sheng et al., 2018). In this study, miR-7a-5p was highly expressed in the rat pituitary gland. This finding indicates that miR-7a-5p plays a potentially vital role in the secretion of hormones and the regulation of sequential production in animals.

According to many studies, the miR-7 family plays different roles in different cancers and has pathological significance in cancer. MiR-7-5p acts as a tumor suppressor in pancreatic ductal adenocarcinoma and suppresses cell proliferation, migration and invasion by targeting SOX18 (Zhu et al., 2018); in MCF-10A mammary epithelial cells, this miRNA suppresses oncogenes by mediating the signaling of hepatocyte growth factor (Jeong et al., 2017). However, miR-7 overexpression in normal fibroblasts (NFs) dramatically increased cancer cell coculture growth rates and migratory activity (Shen et al., 2017). Moreover, the miR-7 family is involved in diverse cellular functions. MiRNA-7a plays a role in Müller glial differentiation via blockade of Notch3 expression (Baba, Aihara & Watanabe, 2015) and alleviates the maintenance of neuropathic pain by regulating neuronal excitability (Sakai et al., 2013). In addition, miR-7 promotes Drosophila wing growth by controlling the Notch signaling pathway (Aparicio, Simoes Da Silva & Busturia, 2015). Importantly, miR-7 may slow Parkinson’s disease progression and regulate proliferation and the mTOR pathway (Titze-de-Almeida & Titze-de-Almeida, 2018; Wang et al., 2013). In 2017, Ahmed et al. (2017) found that normal pituitary development depended on the participation of miR-7a2 and that genetic deletion of miR-7a2 caused infertility. Moreover, the lack of miR-200b and miR-429 exerted the same biological effects, anovulation and infertility (Hasuwa et al., 2013). Accordingly, as a single member of the miR-7 family, miR-7a-5p may regulate pituitary development and reproduction. Based on our results, miR-7a-5p overexpression attenuates FSHb expression and decreases FSH secretion, contributing to the mechanism underlying FSH regulation by miRNAs.

In the past decade, miRNAs have been found in various tissues and organs. However, most mature miRNAs exhibit tissue-specific expression patterns with a precise timing trend that crucially contribute to cell identity and function (Choudhury et al., 2013; Landgraf et al., 2007). In 2002, Lagos-Quintana et al. (2002) examined nine different mice and identified 34 highly tissue-specific novel miRNAs. In 2014, in a study of the pig genome, Martini et al. (2014) predicted species-specific and conserved miRNAs and identified many tissue-specific miRNAs in different tissues. Moreover, some muscle-specific miRNAs, such as miR-1, miR-133a, miR-133b and miR-206, were validated in 2014 (Takeuchi, Sakamoto & Takizawa, 2014). In our study, miR-7a-5p was highly expressed in the brain and pituitary, consistent with the results of an in situ hybridization study (Herzer, Silahtaroglu & Meister, 2012). These data indicate that miR-7a-5p has a potential regulatory function in the pituitary gland and brain.

Gonadotropin FSH, one of the major hormones secreted by the anterior pituitary gland, has a critical role in regulating reproduction (Ulloa-Aguirre et al., 1995). Therefore, elucidating the mechanisms involved in FSH regulation is important. Although many studies have reported that miRNAs can inhibit the secretion of FSH, there are many other factors that influence the secretion of FSH to ensure the growth and development of animals, such as follistatin (Meriggiola et al., 1994), hormones (Dumesic et al., 2009) and single nucleotide polymorphisms (Dai et al., 2009). For example, triiodothyronine differentially modulates FSH synthesis and secretion in male rats, and the Bu-shen-zhu-yun decoction promotes FSH synthesis and secretion. Gonadotropin releasing hormone (GnRH) is a major regulator of FSH secretion, and differential GnRH pulse frequencies and amplitudes affect FSH secretion levels and patterns (Belchetz et al., 1978; Savoy-Moore & Swartz, 1987). Nevertheless, little is known regarding the association of miRNAs with FSH secretion. In 2013, several miRNAs were shown to target the FSHb mRNA 3′UTR after pituitary cells were treated with 100 nM GnRH (Ye et al., 2013). Additionally, the activation of FSH expression is dependent on miR-132/212 (Lannes et al., 2015). In our previous study, miR-186, miR-433 and miR-21-3p were confirmed to regulate FSHb expression and FSH secretion by directly targeting the FSHb 3′UTR (Han et al., 2017; Sheng et al., 2018). Furthermore, in this study, miR-7a-5p had the same effect on FSHb. This study will help improve our understanding of the regulatory functions of miRNAs in the pituitary, enriching our knowledge regarding the mechanism underlying FSH regulation.

Conclusion

Overall, our study demonstrated a role for miR-7a-5p in suppressing FSHb expression and decreasing FSH secretion. These findings provide additional evidence that miRNAs may regulate FSH secretion by directly targeting FSHb.

Supplemental Information

Supplemental Information 1 File S1. Construction of pmiR-FSHb-3’UTR-MUT reporter plasmid.

Click here for additional data file.

Supplemental Information 2 Raw data for Figures 1–4.

Click here for additional data file.

Additional Information and Declarations

Competing Interests

Author Contributions

Animal Ethics

Data Availability

The authors declare that they have no competing interests.

Chang-Jiang Wang performed the experiments, analyzed the data, prepared figures and/or tables, authored or reviewed drafts of the paper, approved the final draft.

Hai-Xiang Guo performed the experiments, analyzed the data, prepared figures and/or tables, authored or reviewed drafts of the paper, approved the final draft.

Dong-Xu Han performed the experiments, analyzed the data, prepared figures and/or tables, authored or reviewed drafts of the paper, approved the final draft.

Ze-Wen Yu performed the experiments, analyzed the data, approved the final draft.

Yi Zheng performed the experiments, analyzed the data, approved the final draft.

Hao Jiang performed the experiments, contributed reagents/materials/analysis tools, approved the final draft.

Yan Gao performed the experiments, contributed reagents/materials/analysis tools, approved the final draft.

Bao Yuan conceived and designed the experiments, approved the final draft.

Jia-Bao Zhang conceived and designed the experiments, approved the final draft.

The following information was supplied relating to ethical approvals (i.e., approving body and any reference numbers):

This experiment was approved by the Institutional Animal Care and Use Committee of Jilin University (201705026).

The following information was supplied regarding data availability:

The raw data are available as a Supplemental File.

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
