# Peer review of "Pituitary tissue-specific miR-7a-5p regulates FSH expression in rat anterior adenohypophyseal cells"

_PeerJ, doi:10.7717/peerj.6458_

## Round 0.1 · original submission · Minor Revisions

The authors should address some minor issues raised by both reviewers to improve the quality of their manuscript.

Reviewer 1 ·

Basic reporting

The manuscript is well written and organized.
A few minor comments are:
1. It would be helpful to readers if the authors provided additional details around the miR-7a-5p mimic and inhibitor that were used.
2. All prior studies around FSH regulation by miRNAs discussed in the Discussion have been conducted only in the context of cell culture. It would interesting if the authors could discuss work, if any, that has been done in vivo (e.g., in rats), or comment on this limitation.

Experimental design

The manuscript presents robust data to support its conclusions.

Validity of the findings

No comment

Reviewer 2 ·

Basic reporting

In this manuscript, Wang CJ et al., aim to demonstrate the role of miR-7a-5p in regulating the expression of follicle-stimulating hormone (FSH). Overall, the manuscript is well structured, and provides evidence to suggest a role for miR-7a-5p in down regulating expression of FSH. A few minor revisions could help strengthen the quality of the manuscript:

There are several issues with figure 2. Firstly, the method used to determine miR-7a-5p expression (qPCR?) is not stated in the figure legend or main text. This also applies to figure 3b & 4a, and needs to be clarified. Secondly, it appears expression in figure 2A was normalized to immature animals (by setting this value to 1). If so, it should be explicitly stated (also applies to figure 3b).

The overall design of the luciferase reporter used in figure 1 isn’t immediately obvious from reading the figure legend. Is only the miRNA seed sequence fused to luciferse, or is a larger fragment of the gene included? Perhaps, adding a cartoon or visual graphic of the overall structure of the reporter construct would be helpful.

In several figures, the precise cell line used is not clearly indicated in the figure legend (e.g. fig. 4). Clearly indicating such information in the figure legend will help improve the readability of the manuscript

There are a few instances of grammatically incorrect sentences. For example, line 158 (“was been acquired”), line 170 (changes in expression over development not “discrepancies”), etc.

Experimental design

In line 167, the authors claim that “mir-7a-5p directly regulates FSHb expression by directly targeting the FSHb gene”, based on a reporter assay where presumably the 3’UTR of FSHb (or a mutated sequence) is place upstream of luciferase reporter. While this proves that the FSHb 3’UTR has a potential binding site for mir-7a-5p, direct targeting cannot be concluded from this experiment. Similarly, the experiment in figure 4b certainly proves that mir-7a-5p expression can regulate FSHb levels, but once again, direct engagement is possible, but not the only explanation. To conclude direct targeting, an in vitro cleveage assay with mir-7a-5p, recombinant AGO2 protein and full-length synthetic FSHb mRNA should be used. If such an experiment is not feasible, the authors should consider adjusting the language to reflect alternate hypothesis.

In the experiment in figure 2A it is unclear what tissues from the immature and mature rats were used in determining miR-7a-5p expression. This is important to define as it also relates to the data presented in figure 2B.

Validity of the findings

The authors find that expression of mir-7a-5p is highest in rat pituitary glands. However, the pituitary is also where FSH expression is highest. Obviously, a direct anti-correlation is unreasonable to expect, but perhaps a brief discussion as to other positive regulators of FSH in the pituitary would help address this point.

Similarly to the above, the expression of mir-7a-5p is also significant in the rat brain. Assuming this isn’t merely a contamination associated with separation of pituitary from other brain tissue, this finding could be interesting as it might suggest a function of FSH in the brain. The authors should consider adding a few sentences (even if speculative) addressing this in the discussion.

---

## Round 0.2 · Minor Revisions

The manuscript has been substantially improved. However, figure 4 legend should be rechecked, and English should be edited by a native English speaker. Please, consider this latter as an important issue.

Reviewer 1 ·

Basic reporting

The authors have made a good faith effort to address both my comments. I do not have any additional feedback.

Experimental design

The authors have made a good faith effort to address both my comments. I do not have any additional feedback.

Validity of the findings

The authors have made a good faith effort to address both my comments. I do not have any additional feedback.

Reviewer 2 ·

Basic reporting

The authors have clearly taken my previous comments in a constructive manner and implemented significant changes through the manuscript, and I commend them for their effort.

My only minor remaining comments are:
1) In response to comment#3, "In several figures, the precise cell line used is not clearly indicated in the figure legend (e.g. fig. 4). Clearly indicating such information in the figure legend will help improve the readability of the manuscript", the authors respond with "I have added a detailed description in the figure legend". Yet I do not see any significant change to figure 4 legend, and this information is still missing from the legend.

2) There are still a few instances of grammatical errors. I would recommend a native English speaker review the manuscript in detail to fix these.

Experimental design

No Comment

Validity of the findings

No Comment

Additional comments

No Comment

---

## Round 0.3 · accepted · Accept

The authors have successfully addressed the reviewers' and editor's comments.

#